# SeaDAG: Semi-autoregressive Diffusion for Conditional Directed Acyclic Graph Generation

## Abstract

We introduce SeaDAG, a semi-autoregressive diffusion model for conditional generation of Directed Acyclic Graphs (DAGs). Considering their inherent layer-wise structure, we simulate layer-wise autoregressive generation by designing different denoising speed for different layers. Unlike conventional autoregressive generation that lacks a global graph structure view, our method maintains a complete graph structure at each diffusion step, enabling operations such as property control that require the full graph structure. Leveraging this capability, we evaluate the DAG properties during training by employing a graph property decoder. We explicitly train the model to learn graph conditioning with a condition loss, which enhances the diffusion model's capacity to generate graphs that are both realistic and aligned with specified properties. We evaluate our method on two representative conditional DAG generation tasks: (1) circuit generation from truth tables, where precise DAG structures are crucial for realizing circuit functionality, and (2) molecule generation based on quantum properties. Our approach demonstrates promising results, generating high-quality and realistic DAGs that closely align with given conditions.

## 1 Introduction

The success of diffusion models in various domains (Dhariwal & Nichol, 2021; Kong et al., 2021) has led to significant interest in their application to graph generation (Vignac et al., 2023; Kong et al., 2023). In this work, we focus on conditional Directed Acyclic Graph (DAG) generation. DAGs are essential and widely used data structures in various domains, including logic synthesis (Li et al., 2022; Liu & Young, 2023; Wang et al., 2023; Pei et al., 2024) and bioinformatics (Zhou & Cai, 2019). Compared to general graphs, DAGs possess an **inherent layer-wise structure with intricate node inter-dependencies** that can significantly influence the overall graph properties. In logic synthesis, for example, minor structural alterations in lower layers can propagate errors to higher layers, resulting in substantial functional differences. Modeling these layer-wise structural features of DAGs requires specially designed model architectures and generation mechanisms (An et al., 2024). Recognizing these challenges, many established DAG synthesis and optimization tools (Mishchenko et al., 2006; Flach et al., 2014; Li et al., 2024b; Wang et al., 2024) employ a sequential synthesis approach, allowing for effective propagation and modeling of localized changes at each step on the global DAG structure.

Recent studies have explored DAG generation using Autoregressive (AR) diffusion models (Li et al., 2024a), owing to their improved efficiency and enhanced ability to model node-edge dependencies (Kong et al., 2023). However, existing approaches generally face the following challenges: (1) Their *strict part-by-part generation* impedes information flow from later components to earlier ones, whereas in DAG structures with complex inter-layer interactions, subsequent layer structure can often influence the message passing in preceding layers (Wu & Qian, 2016). (2) They *lack a global graph structure view* until the final generation step. This limitation is particularly problematic in conditional generation scenarios, where graph properties or functions generally cannot be evaluated without a complete structure (Fang et al., 2022). This incomplete view of conventional AR methods hinders effective property guidance during both training and sampling (Vignac et al., 2023). (3) Many existing works *do not employ explicit condition learning* during training, with con-

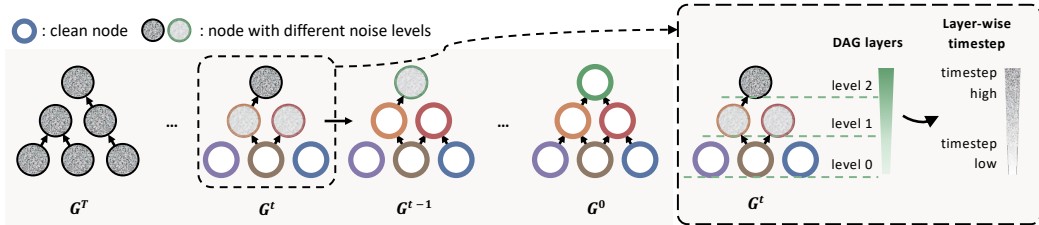

*The figure illustrates the noise levels for nodes only, omitting edges for clarity.*

Figure 1: The overview of proposed layer-wise semi-autoregressive diffusion of SeaDAG. Layers are denoised at different speeds depending on their levels in DAG. A complete graph structure is maintained at very step.

ditional guidance applied only during sampling (An et al., 2024; Vignac et al., 2023). However, our experiments suggest that incorporating explicit graph condition learning during training enables the model to more effectively balance condition satisfaction and graph realism.

To address above issues while preserving the benefits of AR models, we propose **SeaDAG**, a *SEmi-Autoregressive* diffusion-based DAG generation model that enables graph property control, as illustrated in Figure 1. Our approach simulates layer-wise autoregressive generation while being able to output a complete graph at every diffusion step. We achieve this by applying different denoising speeds to nodes and edges within different layers, inspired by recent advances in natural language generation (Wu et al., 2023). Therefore, layers with higher-noise can be generated conditioned on other less-noisy layers.

In summary, SeaDAG offers several key advantages over existing methods:

(1) **SeaDAG fully exploits the hierarchical and layer-wise nature of DAGs**. We design a denoising process that mimics sequential AR generation, enabling the model to effectively capture inter-layer dependencies.

(2) **SeaDAG evolves all layers simultaneously**, albeit at different rates, Unlike the strict part-by-part generation, this simultaneous evolution enables more flexible generation and message passing among layers.

(3) **SeaDAG maintains a complete graph structure at each timestep**, akin to one-shot methods. Leveraging this, we **incorporate explicit condition learning** by employing a property decoder during training to evaluate graph properties. By using a condition loss, we explicitly teach the model to learn the relationship between DAG structures and properties, which helps the model simultaneously satisfy the conditions while producing realistic graph structures.

To demonstrate the broad applicability of our method, we evaluate it on two significant conditional graph generation tasks from distinct domains. First, we address an important challenge in electronic design automation (Liu et al., 2023): circuit generation from truth tables. This task was selected due to the pervasive use of DAGs in circuit design, representing a classic application of DAGs in real-world engineering. Second, to showcase our model's versatility, we tackle molecule generation based on quantum properties, a more general graph generation task. For this application, we convert molecular structures into DAGs using the junction tree representation (Jin et al., 2018). Notably, our model surpasses many specialized molecule generation models. These results collectively show the robustness of our method to produce realistic DAGs that adhere to specified conditions across diverse domains.

## 2 METHODS

### 2.1 PRELIMINARY

**Directed acyclic graph**   A directed acyclic graph with $n$ nodes $\mathcal{V} = \{n_1, n_2 \ldots n_n\}$ can be represented as $G = (\mathbf{X}, \mathbf{E})$. $\mathbf{X} \in \mathbb{R}^{n \times k_x}$ is the node type matrix. Each row $x_i \in \mathbb{R}^{k_x}$ is a one-hot vector encoding the type of node $n_i$ among $k_x$ possible types. $\mathbf{E} \in \mathbb{R}^{n \times n \times k_e}$ is the edge type matrix. Each

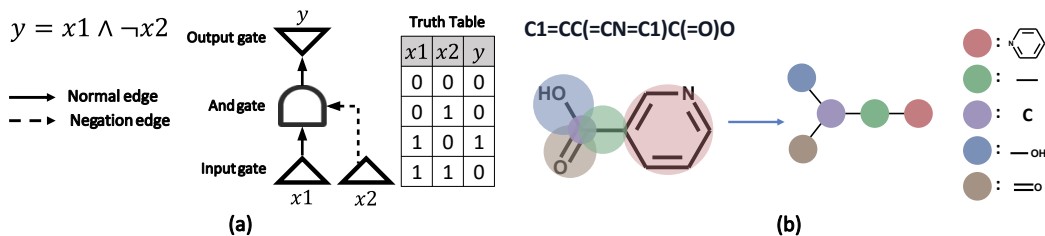

Figure 2: (a) Example AIG and its truth table. (b) Example molecule and its junction tree, where each node represents a chemical substructure. The tree will be further transformed into a DAG.

entry $e_{ij} \in \mathbb{R}^{k_e}$ is a one-hot vector encoding the type of the directed edge from $n_i$ to $n_j$ among $k_e$ possible types. In this work, for a directed edge $e_{ij}$, we refer to $n_i$ as the child node and $n_j$ as the parent node. The $i$th row of $\mathbf{E}$ encodes all parents of $n_i$, denoted as $\text{Pa}(n_i)$. The $j$th column encodes all children of $n_j$, denoted as $\text{Ch}(n_j)$. We define leaf nodes as nodes without children, while root nodes are those without parents. The level of node $n_i$ is defined as the length of the longest directed path from any leaf node to $n_i$ (Bezek, 2016):

$$\text{level}(n_i) = \max_{n_j \in \text{Ch}(n_i)} \text{level}(n_j) + 1, \text{if } n_i \text{ is not a leaf node,} \tag{1}$$

$$\text{level}(n_i) = 0, \text{if } n_i \text{ is a leaf node.} \tag{2}$$

**DAG representation of circuits**  The And-Inverter Graph (AIG) is a powerful representation of logic circuits. In an AIG, leaf nodes carry input signals to the circuit, while root nodes produce the circuit's output signals. Intermediate nodes are computing units that perform logical conjunction (AND operation) on two binary inputs $a, b \in \{0, 1\}$, outputting $a \wedge b$. Edge $e_{ij}$ from node $n_i$ to $n_j$ indicates that the output of $n_i$ serves as an input to $n_j$. Edges can optionally perform logical negation (NOT operation) on the signal they carry. This representation naturally forms a DAG structure. The structure of an AIG uniquely defines its functionality, which can be represented as a canonical truth table mapping all possible combinations of input signals to their corresponding output signals. Figure 2(a) illustrates a simple AIG and its truth table.

This representation is particularly useful because any combinational logic circuits can be expressed using only AND and NOT operations, making AIG a universal and efficient model for circuit analysis and synthesis. In this work, we generate AIGs that can realize a given truth table functionality. For node types, we have three category ($k_x = 3$): input gates, AND gates and output gates. For edges, we define three types ($k_e = 3$): non-existing edges representing the absence of a connection, normal edges for direct connections, and negation edges that perform a logical NOT operation.

**DAG representation of molecules**  We adopt the approach of Jin et al. (2018) to convert molecules into junction trees. This method first extracts valid chemical substructures, e.g. rings, bonds and individual atoms, from the training set. The vocabulary size is the number of node types $k_x$. Each molecule is then converted into a tree structure representing the scaffold of substructures. We transform this tree into a DAG by designating the tree root as the DAG root and converting tree edges to directed edges from children to parents. Edges have two types ($k_e = 2$): absent connections and existing connections. In this work, we generate junction trees of molecules from their properties. Junction trees are subsequently assembled into complete molecules for evaluation. Figure 2(b) shows a molecule and its junction tree. For more details on the construction of junction trees and their conversion to molecules, we refer the reader to Jin et al. (2018).

### 2.2 SEMI-AUTOREGRESSIVE DIFFUSION FOR DAG

In this section, we introduce the proposed semi-autoregressive diffusion generation for DAG. This approach considers the hierarchical nature of DAGs while maintaining a complete graph structure at every step of the sampling process. The training pipeline of SeaDAG is illustrated in Figure 3.

**Discrete graph denoising diffusion**  The forward diffusion process independently adds noise to each node $x_i$ and each edge $e_{ij}$ from timestep 0 to $T$, where $T$ is the maximum diffusion step.

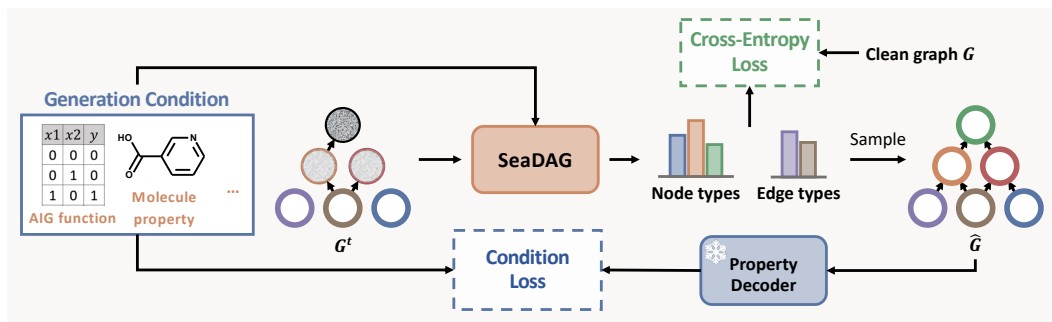

Figure 3: Training pipeline. SeaDAG predicts the node and edge type distribution in the clean graph. Apart from the cross-entropy loss, we employ a condition loss to incorporate explicit condition learning during training.

The forward process is defined using transition matrices: $[Q_X^t]_{ij} = q(x^t = \mathbf{e}_j | x^{t-1} = \mathbf{e}_i)$ and $[Q_E^t]_{ij} = q(e^t = \mathbf{e}_j | e^{t-1} = \mathbf{e}_i)$. Here, $\mathbf{e}_i$ denotes a one-hot vector with 1 in its $i$th position. Consequently, the node and edge types at time $t$ can be sampled from the following distributions:

$$q(x_i^t | x_i^{t-1}) = Q_X^{t'} x_i^{t-1}, \; q(e_{ij}^t | e_{ij}^{t-1}) = Q_E^{t'} e_{ij}^{t-1}, \tag{3}$$

$$q(x_i^t | x_i) = \bar{Q}_X^{t'} x_i, \; q(e_{ij}^t | e_{ij}) = \bar{Q}_E^{t'} e_{ij}, \tag{4}$$

where $\bar{Q}_X^t = Q_X^1 \ldots Q_X^t$ and $\bar{Q}_E^t = Q_E^1 \ldots Q_E^t$. Following Vignac et al. (2023), we use marginal distribution to define transition matrices: $Q_X^t = \alpha_t \mathbf{I} + (1 - \alpha_t) \mathbf{1} \mathbf{m_X}, Q_E^t = \alpha_t \mathbf{I} + (1 - \alpha_t) \mathbf{1} \mathbf{m_E}$, where $\mathbf{m_X}, \mathbf{m_E}$ are marginal distributions of node and edge types. We use cosine noise schedule following Nichol & Dhariwal (2021).

**Semi-Autoregressive diffusion**  To leverage the inherent layer-wise structure of DAGs and the dependency-modeling advantages of sequential generation, we introduce different diffusion speed for different layers. We define the timestep $t \in [0, T]$ as the global timestep. We denote the normalized node level as $l_i = \text{level}(n_i) / \max_{n_j \in \mathcal{V}} \text{level}(n_j)$. We then design a function $\mathcal{T} : [0, T] \times [0, 1] \to [0, T]$ that maps the global timestep $t$ and normalized level $l_i$ to a node-specific local timestep $\tau_i^t = \mathcal{T}(t, l_i)$ for node $n_i$, or an edge-specific local timestep $\tau_{ij}^t = \mathcal{T}(t, l_j)$ for edge $e_{ij}$. By designing $\mathcal{T}$ such that $\mathcal{T}(t, l_i) >= \mathcal{T}(t, l_j)$ when $l_i > l_j$, we assign larger timesteps to nodes at higher levels and edges pointing to higher levels. This configuration results in a bottom-up generation process, where layers at the bottom of the DAG are denoised at a higher speed. Conversely, we can achieve top-down generation by reversing this relationship. In our experiments, we implement $\mathcal{T}$ as:

$$\text{Bottom up generation: } \mathcal{T}(t, l) = \text{clip}(\frac{T}{T - \beta(1 - l)}(t - \beta(1 - l)), 0, T) \tag{5}$$

$$\text{Top down generation: } \mathcal{T}(t, l) = \text{clip}(\frac{T}{T - \beta l}(t - \beta l), 0, T), \tag{6}$$

where $\beta$ is a hyperparameter and function $\text{clip}(x, a, b)$ clips input $x$ to $[a, b]$. Empirically, we adopt a bottom-up generation approach for AIGs and a top-down generation approach for molecules.

During training, we sample a random global timestep $t$ from $[0, T]$ and sample $G^t = (\mathbf{X}^t, \mathbf{E}^t)$ from the distribution:

$$q(G^t | G) = \prod_{1 \le i \le n} q(x_i^{\tau_i^t} | G) \prod_{1 \le i, j \le n} q(e_{ij}^{\tau_{ij}^t} | G) \tag{7}$$

$$q(x_i^{\tau_i^t} | G) = \bar{Q}_X^{\tau_i^t \,\prime} x_i, \; q(e_{ij}^{\tau_{ij}^t} | G) = \bar{Q}_E^{\tau_{ij}^t \,\prime} e_{ij}. \tag{8}$$

We train a network $f_\theta$ to predict the distribution of real graph $G$ from the noisy graph $G^t$: $f_\theta(G^t) = (p_\theta(\mathbf{X}), p_\theta(\mathbf{E}))$. Specifically, we extend the graph transformer architecture from Dwivedi & Bresson

(2020). We employ cross-entropy loss between the predicted distribution and ground truth $G$ to compute the graph reconstruction loss $\mathcal{L}_{graph}$:

$$\mathcal{L}_{graph}(\theta) = \sum_{1 \leq i \leq n} \text{Cross-Entropy}(x_i, p_\theta(x_i)) + \sum_{1 \leq i,j \leq n} \text{Cross-Entropy}(e_{ij}, p_\theta(e_{ij})). \quad (9)$$

During inference, we can use the predicted distribution to sample $G^{t-1}$ from $G^t$:

$$p_\theta(G^{t-1}|G^t) = \prod_{1 \leq i \leq n} p_\theta(x_i^{\tau_i^{t-1}}|G^t) \prod_{1 \leq i,j \leq n} p_\theta(e_{ij}^{\tau_{ij}^{t-1}}|G^t), \quad (10)$$

where

$$p_\theta(x_i^{\tau_i^{t-1}}|G^t) = \sum_{k \in \{1...k_x\}} p_\theta(x_i^{\tau_i^{t-1}}|x_i = \mathbf{e}_k, G^t)p_\theta(x_i = \mathbf{e}_k|G^t) \quad (11)$$

$$p_\theta(x_i^{\tau_i^{t-1}}|x_i, G^t) = p_\theta(x_i^{\tau_i^{t-1}}|x_i, x_i^{\tau_i^t}) = \frac{p(x_i^{\tau_i^t}|x_i^{\tau_i^{t-1}}, x_i)q(x_i^{\tau_i^{t-1}}|x_i)}{q(x_i^{\tau_i^t}|x_i)} \quad (12)$$

$$p(x_i^{\tau_i^t}|x_i^{\tau_i^{t-1}}, x_i) = \sum_{x_i^{\tau_i^{t-1}+1}} \cdots \sum_{x_i^{\tau_i^{t-2}}} \sum_{x_i^{\tau_i^{t-1}}} \prod_{\tau = \tau_i^{t-1}+1}^{\tau_i^t} q(x_i^\tau|x_i^{\tau-1}), \quad (13)$$

and $p_\theta(e_{ij}^{\tau_{ij}^{t-1}}|G^t)$ can be similarly computed. Detailed computation is provided in Appendix C.

**Sampling from SeaDAG**  Computing node and edge-specific timesteps requires the node levels in the clean DAG. During training, we have access to the ground truth clean graph $G$ and could directly compute node levels. During sampling, where we start from a random graph, we determine its hierarchical structure by sampling the number of levels and their sizes from distributions observed in the training data. We can then apply Equation 10 to perform backward denoising and generate the final DAG. Detailed sampling algorithm is provided in Appendix D.1.

**Model equivariance**  We can prove that our method is permutation equivariant since the utilized graph generation model is equivariant and the loss is invariant to node permutations. Detailed proofs are provided in Appendix C.

## 2.3 CONDITIONAL GENERATION

Our approach incorporates explicit condition learning by employing a property decoder to compute a condition loss during training. This section details the implementation of these key components.

### 2.3.1 CONDITION LOSS

To incorporate graph properties as generation conditions, we extend our network to accept an additional condition input $cond$: $f_\theta(G^t, cond) = (p_\theta(\mathbf{X}), p_\theta(\mathbf{E}))$. For AIG generation from truth tables, we concatenate truth table vectors with other node features. Each column of binary values in the truth table serves as an additional feature for the corresponding node. Since the condition truth table only provides signals for the input and output gates, we use all zero signals for the AND gates. In molecule generation from quantum property, we concatenate the desired molecule property with the graph level features.

To enhance the network's ability to generate graphs that meet given conditions, we introduce a differentiable graph property decoding module $\phi$. This module decodes the graph property from a clean graph. The implementation of $\phi$ varies based on the application: (1) For AIG generation, we simulate continuous circuit output using softmax-choice wiring, where each gate's input is a soft distribution over candidate inputs rather than a hard selection. (2) For applications where the graph property is not directly computable, such as molecule generation, we employ a pre-trained prediction model as the decoding module. Detailed implementations of $\phi$ can be found in Appendix B.3. During training, we sample a predicted clean graph $\hat{G}$ from the predicted distribution $(p_\theta(\mathbf{X}), p_\theta(\mathbf{E}))$

using Gumbel-Softmax and decode its property using $\phi$. Then we compute the condition loss $\mathcal{L}_{cond}$ between ground truth property condition $cond$ and the decoded property:

$$\mathcal{L}_{cond}(\theta) = \text{LossFunction}(cond, \phi(\hat{G})), \tag{14}$$

where LossFunction is Binary Cross Entropy for comparing two binary truth tables and Mean Squared Error for molecule property. The final loss is the combination of $\mathcal{L}_{graph}$ and $\mathcal{L}_{cond}$:

$$\mathcal{L}(\theta) = \mathcal{L}_{graph}(\theta) + \lambda\mathcal{L}_{cond}(\theta), \tag{15}$$

where $\lambda$ is a hyperparameter. Since the $\mathcal{L}_{cond}$ is also invariant, the total loss is still invariant to permutation. The training algorithm of SeaDAG is presented in Algorithm 1.

---

**Algorithm 1** SeaDAG training algorithm

---

**Input:** DAG dataset $\{(G = (\mathbf{X}, \mathbf{E}), cond)\}$, maximum timestep T, function $\mathcal{T}$ to map global timestep $t$ and node level $l_i$ to local timestep, property decoder $\phi$
**Output:** Optimized model parameter $\theta$
  **while** not converged **do**
    Sample $(G, cond)$ and global timestep $t \in [1, T]$
    Compute **node and edge specific local timestep**

$$\tau_i^t \leftarrow \mathcal{T}(t, l_i), \tau_{ij}^t \leftarrow \mathcal{T}(t, l_j)$$

    Sample $G^t \sim q(G^t|G)$ using $\tau_i^t, \tau_{ij}^t$ based on Equation 4
    $p_\theta(\mathbf{X}), p_\theta(\mathbf{E}) \leftarrow f_\theta(G^t, cond)$
    $\mathcal{L}_{graph} \leftarrow \text{Cross-Entropy}(\mathbf{X}, p_\theta(\mathbf{X})) + \text{Cross-Entropy}(\mathbf{E}, p_\theta(\mathbf{E}))$
    Sample $\hat{G} \sim (p_\theta(\mathbf{X}), p_\theta(\mathbf{E}))$ for **condition loss** calculation

$$\mathcal{L}_{cond} \leftarrow \text{LossFunction}(cond, \phi(\hat{G}))$$

    Optimize $\theta$ to minimize $\mathcal{L}_{graph} + \lambda\mathcal{L}_{cond}$
  **end while**

---

### 2.3.2 INTEGRATING CONDITON LEARNING INTO TRAINING

Many AR or one-shot approaches to conditional graph generation train an unconditional model and incorporate conditional guidance only during sampling (Vignac et al., 2023; Kong et al., 2023). However, we argue that this separation of condition learning from training hinders the model's ability to balance condition satisfaction with graph realism. Our experiment results suggest that while these methods can generate realistic graphs in unconditional mode, they struggle to maintain graph quality after applying conditional guidance during sampling. In contrast, by integrating conditional learning into the training process, our method SeaDAG achieves a more effective balance between adhering to conditions and producing plausible graph structures.

## 3 RELATED WORKS

Different diffusion mechanisms have been proposed for graph generation. One-shot generation models apply noise addition and denoising processes across the entire graph structure simultaneously, predicting all nodes and edges at each timestep (Yan et al., 2024; Vignac et al., 2023). In contrast, AR diffusion models generate the graph sequentially, either producing one part of the graph at each diffusion step (Kong et al., 2023) or having a separate denoising process for each part (Zhao et al., 2024). These AR approaches offer advantages in generation efficiency and are better at modeling dependencies between nodes and edges by allowing each step to condition on previously generated parts. (Kong et al., 2023). However, they face challenges such as sensitivity to node ordering (You et al., 2018b) and the production of only partial, incomplete graph structures during the sampling process. This limitation precludes operations that require the entire graph structure, such as validity checks and property guidance (Vignac et al., 2023; Yu et al., 2023). Latent diffusion models have also been explored for graph generation (Zhou et al., 2024), with studies indicating their potential to enhance performance in 3D graph generation tasks (You et al., 2023).

## 4 EXPERIMENTS

In our experiments, we focus on evaluating two key aspects of the methods: (1) **Conditional generation**: We assess whether the methods can generate graphs that satisfy the given condition. (2) **Graph quality**: We evaluate whether the generated graphs are realistic and resemble real graphs in their distribution.

### 4.1 DATA AND BASELINES

**AIG dataset**  We generate a dataset of random AIGs with 8 inputs, 2 outputs and a maximum of 32 gates. For each AIG, we compute its corresponding truth table. The dataset comprises 12950 AIGs in the training set, 1850 in the validation set, and 256 in the test set. The generalization experiments on AIGs with more than 8 input and 2 outputs are available in Appendix A.2.

**Molecule dataset**  For molecule generation, we evaluate SeaDAG on the standard QM9 benchmark (Wu et al., 2017), which contains molecules with up to 9 heavy atoms. We adopt the standard dataset split, utilizing 100k graphs for training and 20k for validation. We generate 10k molecules for evaluation. We further conduct a molecule optimization experiment using the ZINC dataset (Irwin et al., 2012), which has 219k molecules in training set and 24k molecules for validation. More dataset statistics are provided in Appendix E.1.

**Baselines**  We evaluate SeaDAG against several recent graph generation diffusion models as well as state-of-the-art molecule generation models. For one-shot graph diffusion baselines, we compare SeaDAG with SwinGNN (Yan et al., 2024), EDP-GNN (Niu et al., 2020), GDSS (Jo et al., 2022) and DiGress (Vignac et al., 2023). For AR graph diffusion baselines, we compare SeaDAG with Pard (Zhao et al., 2024) and GRAPHARM (Kong et al., 2023). For graph diffusion baselines that operate in latent space, we compare SeaDAG with LDM-3DG (You et al., 2023) and EDM (Hoogeboom et al., 2022). For molecule generation baselines, we compare SeaDAG with GraphAF (Shi et al., 2020), GraphDF (Luo et al., 2021), MoFlow (Zang & Wang, 2020), GraphEBM (Liu et al., 2021) and SPECTRE (Martinkus et al., 2022). Detailed implementations for baselines are provided in Appendix E.3.

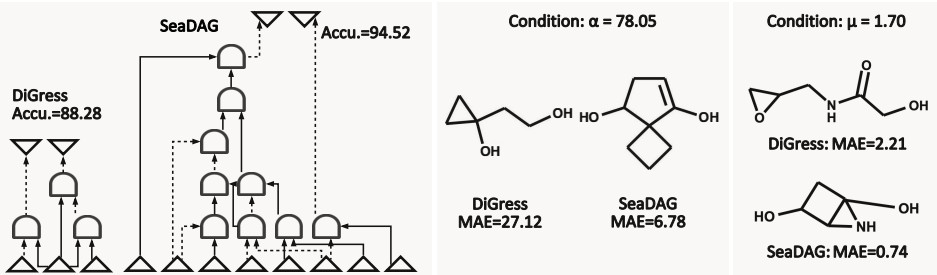

Figure 4: Sampled AIG and molecules by SeaDAG and the one-shot diffusion baseline DiGress.

### 4.2 CONDITIONAL GENERATION

**Conditional AIG generation**  Table 1 presents the evaluation results for AIG generation. To achieve conditional generation, we extend baseline models to accept truth tables as additional input. We report two metrics: *Validity*, which means the percentage of generated AIGs that are structurally valid, and function *Accuracy*, which means the bit-wise accuracy between the condition truth

Table 1: AIG generation. The best results are highlighted in bold and the second-best results are underlined.

| Model | Class | Validity↑ | Accuracy↑ |
|---|---|---|---|
| SwinGNN | One-shot | 88.29 | 57.40 |
| Pard | Autoreg. | 78.86 | 75.78 |
| DiGress | One-shot | 43.21 | 82.07 |
| SeaDAG | Semi-Autoreg. | **92.38** | **89.25** |

table and the truth table of the generated AIGs. SeaDAG significantly outperforms baseline models on both metrics.

Table 2: Conditional generation evaluation on QM9. We report the absolute error between the target and oracle-predicted properties (You et al., 2023). The best results are highlighted in bold and the second-best results are underlined.

| Model | $\alpha$ | HOMO | LUMO | Gap | $\mu$ |
|---|---|---|---|---|---|
| Random | 41.00 | 103.30 | 121.83 | 193.36 | 8.40 |
| EDM | 20.15 | 158.70 | 166.20 | 287.00 | 7.01 |
| LDM-3DG | 15.56 | 54.62 | **63.08** | 107.14 | 6.33 |
| LDM-3DG-GSSL | 16.43 | 55.03 | 66.53 | 113.15 | 9.22 |
| DiGress | 9.23 | 31.98 | 105.06 | 90.57 | 1.49 |
| SeaDAG | **8.85** | **30.91** | 103.03 | **89.70** | **1.33** |

**Conditional molecule generation** We evaluate conditional molecule generation across five molecule properties: polarizability $\alpha$ (Bohr$^3$), Highest Occupied Molecular Orbital (HOMO) energy (meV), Lowest Unoccupied Molecular Orbital (LUMO) energy (meV), Gap between HOMO and LUMO (meV), and dipole moment $\mu$ (D). Table 2 presents the Mean Absolute Error (MAE) between the target molecular properties and properties of the generated molecules. The random baseline represents the MAE of randomly sampled molecules. SeaDAG achieves the lowest MAE in four out of five property conditions, which underscores SeaDAG's ability in generating molecules with desired properties.

## 4.3 GRAPH QUALITY EVALUATION

In addition to evaluating the methods' ability to meet the conditions, we also assess the realism of the generated graphs and how closely they resemble the real graph distribution.

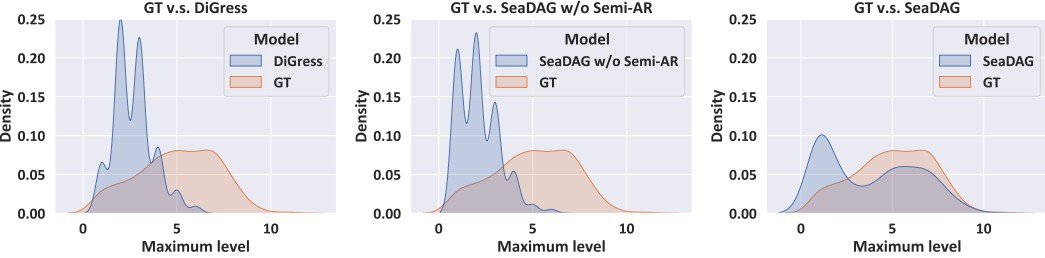

Figure 5: Distribution of maximum levels in generated AIGs. SeaDAG with semi-autoregressive diffusion generates AIGs with maximum levels similar to ground truth AIGs, while other two one-shot methods produce significantly shallower AIGs.

**Semi-autoregressive diffusion enhances AIG realism.** We analyze the quality of generated AIGs from two aspects: (1) the AIG Validity, which is reported in Table 1, and (2) the distribution of maximum levels in the generated AIGs. Figure 5 illustrates a comparison between our proposed SeaDAG model, its one-shot variant (trained without layer-wise semi-autoregressive generation), and the one-shot diffusion model DiGress. While DiGress and the one-shot SeaDAG tend to produce shallow AIGs with significantly fewer levels than the ground truth, SeaDAG with semi-autoregressive diffusion achieves a maximum level distribution closely resembling that of the ground truth AIGs. Figure 6 presents a case study where DiGress and SeaDAG generate an AIG conditioned on the same truth table. SeaDAG is able to generate more complex AIGs with a greater number of levels and obtains higher function accuracy. These observations indicate that semi-autoregressive generation can help the model to construct more complex AIGs that are more similar to ground truth ones and therefore more capable of realizing the given truth tables.

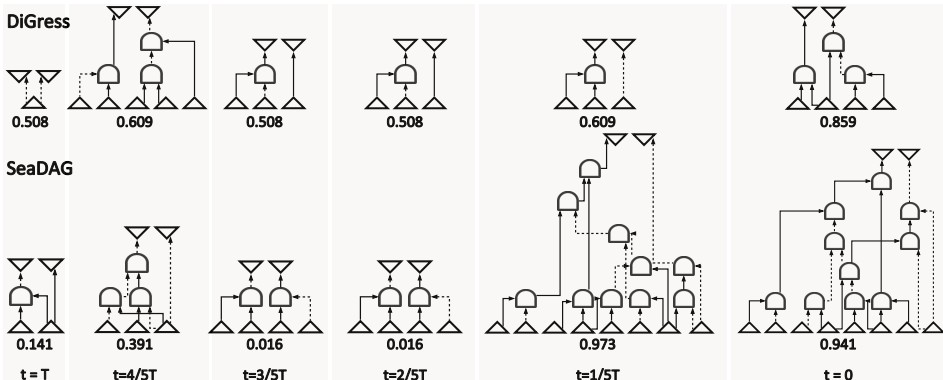

Figure 6: Comparison of a sampled AIG and the function accuracy during the sampling process by DiGress and SeaDAG from the same truth table. SeaDAG is capable of generating AIGs with greater level and complexity, achieving higher function accuracy.

**SeaDAG can balance molecule quality and condition satisfaction.** Table 3 presents the evaluation of the quality of generated molecules. We employ several metrics to assess the plausibility of molecules: **Validity** is the fraction of valid molecules. Neighborhood subgraph pairwise distance kernel (**NSPDK**) MMD (Costa & Grave, 2010) computes the MMD between the generated molecules and the test set, which takes into account both the node and edge features for evaluation. Fréchet ChemNet Distance (**FCD**) is the difference between training set and generated molecules in distributions of last layer activations of ChemNet. Additionally, we also report Uniqueness, which is the fraction of the unique and valid molecules, and Novelty, which is the fraction of unique and valid molecules not present in the training set. Note that for QM9 dataset, a higher Novelty score does not indicate better performance, but rather suggests a deviation from the dataset's distribution, as QM9 comprehensively enumerates molecules within specific constraints (Vignac & Frossard, 2022). We evaluate models in two categories: **Unconditional group**, where molecules are generated without constraints, and **Conditional group**, where generation is guided by one of the five aforementioned properties and the results represent the average of five separate evaluations.

Table 3: Molecule quality evaluation on QM9. Other conditional models, such as LDM-3DG and DiGress, exhibit a decline in quality compared to their unconditional counterparts. In contrast, SeaDAG maintains quality comparable to the best models in the unconditional group. Note that higher novelty on QM9 datasets indicates deviation from the training distribution rather than better performance (Vignac & Frossard, 2022).

| Generation Mode | Model | Validity↑ | NSPDK↓ | FCD↓ | Unique↑ | Novelty |
|---|---|---|---|---|---|---|
| Unconditional | GraphAF | 74.43 | 0.020 | 5.27 | 88.64 | 86.59 |
| | GraphDF | 93.88 | 0.064 | 10.93 | 98.58 | 98.54 |
| | MoFlow | 91.36 | 0.017 | 4.47 | 98.65 | 94.72 |
| | EDP-GNN | 47.52 | 0.005 | 2.68 | **99.25** | 86.58 |
| | GraphEBM | 8.22 | 0.030 | 6.14 | 97.90 | 97.01 |
| | SPECTRE | 87.3 | 0.163 | 47.96 | 35.70 | 97.28 |
| | GDSS | 95.72 | 0.003 | 2.90 | 98.46 | 86.27 |
| | GRAPHARM | 90.25 | 0.002 | 1.22 | 95.62 | 70.39 |
| | LDM-3DG | **100.0** | 0.009 | 2.44 | 97.57 | 89.89 |
| | DiGress | 99.00 | **0.0005** | **0.36** | 96.66 | 33.40 |
| Conditional | LDM-3DG | **100.0** | 0.018 | 4.72 | 81.09 | 92.03 |
| | DiGress | 99.28 | **0.0004** | 0.72 | **96.34** | 47.21 |
| | SeaDAG | **100.0** | 0.002 | **0.36** | 93.01 | 55.15 |

For the baselines in the Conditional group, LDM-3DG (You et al., 2023) achieves conditional generation by giving models additional property inputs and DiGress (Vignac et al., 2023) applys conditional guidance only during sampling stage. Notably, the conditional variants of LDM-3DG and DiGress exhibit up to 100% deterioration in NSPDK and FCD metrics compared to their unconditional counterparts. In contrast, SeaDAG achieves the best FCD scores across both conditional and unconditional groups, while its NSPDK performance is comparable to the best results in both groups.

## 4.4 MOLECULE OPTIMIZATION VIA CONDITIONAL GENERATION

Table 4: Molecule Optimization on ZINC. We report the best property scores achieved by each method. While other methods use optimization algorithms, SeaDAG is solely trained for conditional generation, yet it produces molecules with properties comparable to those of other methods.

| Model | Optimization Algo. | Penalized logP | | | QED | | |
|---|---|---|---|---|---|---|---|
| | | 1st | 2nd | 3rd | 1st | 2nd | 3rd |
| Train set | N.A. | 4.52 | 4.30 | 4.23 | 0.948 | 0.948 | 0.948 |
| ORGAN | Reinforcement Learning | 3.63 | 3.49 | 3.44 | 0.896 | 0.824 | 0.820 |
| JT-VAE | Bayesian Optimization | 5.30 | 4.93 | 4.49 | 0.925 | 0.911 | 0.910 |
| GCPN | Policy Gradient | 7.98 | 7.85 | 7.80 | 0.948 | 0.947 | 0.946 |
| MolecularRNN | Policy Gradient | **10.34** | **10.19** | **10.14** | 0.948 | **0.948** | **0.947** |
| SeaDAG | N.A. | 8.52 | 8.35 | 8.33 | **0.948** | 0.947 | 0.946 |

We demonstrate a practical application of our conditional DAG generation in the domain of molecule optimization. On the ZINC dataset, we train SeaDAG to conditionally generate molecules based on two properties: penalized logP and Quantitative Estimate of Druglikeness (QED), which are two common target properties for molecule optimization (Popova et al., 2019). We then sample from SeaDAG with target property scores. Detailed implementations could be found in Appendix E.6. Without using explicit optimization techniques, SeaDAG achieves top property scores comparable to several optimization-based baselines (Guimaraes et al., 2017; Jin et al., 2018; You et al., 2018a) as shown in Table 4. Notably, for the penalized logP property, SeaDAG attains scores surpassing the highest values observed in the training set. This suggests SeaDAG's capacity to extrapolate beyond the training distribution by effectively learning the intrinsic relationships between molecular structure and associated properties.

## 5 CONCLUSION

In this paper, we introduced SeaDAG, a semi-autoregressive diffusion model for conditional DAG generation. Our approach demonstrates significant improvements in generating graphs that are realistic and realize given conditions. Future research could focus on enhancing the method's efficiency to match that of fully autoregressive models.

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

## A  MORE EXPERIMENT RESULTS

### A.1  MCTS-BASED REFINEMENT FOR CONDITION ALIGNMENT

We implement an MCTS-based post-processing step to enhance the condition properties of the DAGs generated by the diffusion model.

Table 5: Conditional generation results of our proposed MCTS-based DAG structure refinement.

| Model | AIG - Accuracy↑ | Molecule - MAE↓ | | | | |
|---|---|---|---|---|---|---|
| | | $\alpha$ | HOMO | LUMO | Gap | $\mu$ |
| SeaDAG | 89.25 | 8.47 | 32.47 | 105.16 | 89.83 | 1.39 |
| SeaDAG + MCTS | 94.76 | 7.11 | 21.32 | 89.25 | 77.79 | 0.99 |

**State representation**  Each state is a DAG structure $G = (\mathbf{X}, \mathbf{E})$.

**Action space and transition model**   The action space is defined as a set of random graph edits. An action, i.e. a graph edit, for an AIG or a molecule junction tree is defined as follows:

- To sample an action for an AIG, we first randomly sample a gate from the AND gates and output gates. We then change the input gates (i.e. the children) of the chosen gate to nodes randomly sampled from possible candidate input gates. Specifically, an AND gate has two inputs and a output gate only requires one input. The candidate gates are the set of gate nodes with lower levels. Finally, for the new edges from the new inputs to the chosen gate, we determine their types by sampling from normal type and negation type.

- To sample an action for a molecule junction tree, we first randomly select a node from the junction tree. Then, with a probability of 0.5, we either modify the node's type or alter its parent node. In the case of type modification, we randomly assign a new type to the selected node. In the case of changing parent, we randomly select a new parent from the nodes at higher levels in the tree.

Applying an action to a state is to modify the DAG structure using the edit defined by the action and result in a new DAG structure. Our design of the action space ensures that the resulting DAG is always valid.

**Reward function**   We employ the property decoder $\phi$ to compute the reward function for a state.

- For AIGs, we aim to maximize the function accuracy. Therefore, we use the truth table decoder $\phi$ to decode the output signals of the AIG and compute the accuracy as reward.

- For molecules, we aim to minimize the MAE between the molecule property and the target property. Therefore, we employ $\phi$ to predict the molecule property from the junction tree and compute the negative of MAE loss as reward.

**MCTS Algorithm Structure**   Since the action space, namely the set of possible edits to a DAG, is finite but is still very large, we employ the progressive widening strategy (Coulom, 2007; Chaslot et al., 2008) to balance adding new children and selecting among existing children. We employ UCB selection strategy (Auer et al., 2002) to select the best child. In the simulation phase, we employ a random action sampling strategy until reaching the predefined maximum depth limit. Upon reaching this limit, we evaluate the reward of the terminal state and back-propagated it through the tree. We conduct 500 such simulations for each decision point. After these simulations, we select the best child node as the next state. This process is repeated for 50 steps for each DAG.

After the MCTS refinement, we evaluate the resulting DAG structure. If it performs worse than the original DAG, we reject the refinement and retain the original structure.

We evaluate the performance of our MCTS-based DAG structure refinement in Table 5. MCTS is applied to 256 generated AIGs and 600 generated molecule junction trees for each of the five properties, with the aim to improve the function accuracy and property MAE respectively. The MCTS refinement significantly improves performance across all metrics on both tasks.

A.2   MODEL GENERALIZATION

We evaluate SeaDAG's generalization capacity by testing its ability to generate AIGs with varying numbers of input and output gates, as well as diverse total gate counts. Our model uses truth table columns as node-level features, with a fixed length parameter of 256 ($2^8$) rows corresponding to the 8 input gates in the training AIGs. To accommodate AIGs with $N_{\text{input}} > 8$, we randomly sample 256 rows from the $2^{N_{\text{input}}}$ rows of the full truth table. For AIGs with $N_{\text{input}} < 8$, we pad the truth table to 256 rows by randomly duplicating $256 - 2^{N_{\text{input}}}$ rows. Then the truth table can be concatenated to node features and the AIG can be generated in the same way. Figure 7 plots the function accuracy of SeaDAG on AIGs with different number of input gates, output gates and total gates. Although it is only trained on AIGs with 8 input gates and 2 output gates, SeaDAG demonstrates robust performance on AIGs with very different configurations from the training set. Meanwhile, it maintains stable performance across a diverse range of AIG sizes.

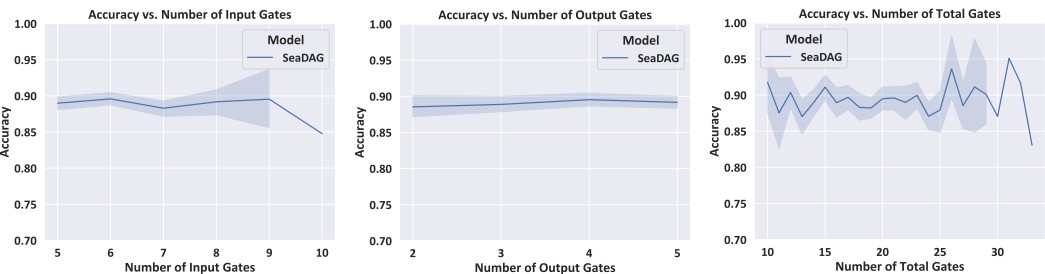

Figure 7: Generalization capacity of SeaDAG across diverse AIG configurations. SeaDAG demonstrates robust generalization to AIGs with input and output gate numbers unseen during training, while maintaining stable performance across a diverse range of AIG sizes.

### A.3 ABLATION STUDY

We conduct ablation study on the two key elements in our method: the condition loss and the semi-autoregressive diffusion scheme.

Table 6: Ablation study results for AIG generation. Note that Accuracy is calculated after correcting the invalid AIG structures.

| Condition Loss | Semi-Autoreg. | Validity↑ | Accuracy↑ |
|---|---|---|---|
| | ✔ | **97.34** | 78.06 |
| ✔ | | 45.25 | **89.84** |
| ✔ | ✔ | 92.38 | 89.25 |

As shown in Table 6, incorporating condition loss during training significantly improves the function accuracy for AIG generation. This suggests that merely providing the condition as an additional input to the model may be insufficient. The condition loss appears to enhance the model's ability to learn the relationship between DAG structure and its property conditions. Conversely, the model without semi-autoregressive generation performs poorly in terms of graph validity (note that we correct invalid AIG graphs when calculating Accuracy). This finding further supports our argument in Section 4.3 that semi-autoregressive diffusion helps the model learn structural features and dependencies within DAGs, leading to the generation of higher quality and more realistic DAG structures.

Table 7 presents the ablation study results for molecule generation. Consistent with our previous findings, the model trained without semi-autoregressive generation shows significant performance degradation across molecule quality metrics. Interestingly, the impact of condition loss on condition satisfaction metrics appears less pronounced in molecule generation compared to AIG generation. We hypothesize that this discrepancy stems from the nature of the property decoders: for junction trees, we employ trained models that may introduce prediction errors, whereas for AIGs, we use precise logic operations. Consequently, the guidance provided by the molecule property decoder may not be as effective as its AIG counterpart. Nonetheless, we can still observe improvements in the MAE metric compared to the ablated models and baselines.

Table 7: Ablation study results for molecule generation on QM9.

| Condition Loss | Semi-Autoreg. | Molecule Quality | | | | | Condition MAE↓ | |
|---|---|---|---|---|---|---|---|---|
| | | Validity↑ | NSPDK↓ | FCD↓ | Unique | Novelty | $\alpha$ | HOMO |
| | ✔ | 100.0 | 0.002 | 0.32 | 93.60 | 54.85 | 9.42 | 31.30 |
| ✔ | | 100.0 | 0.018 | 1.98 | 68.75 | 61.47 | 21.06 | 33.33 |
| ✔ | ✔ | 100.0 | 0.002 | 0.36 | 93.01 | 55.15 | 8.85 | 30.91 |

# B  SeaDAG Parameterization and Training

## B.1  Denoising network implementation

We extend the model employ in Dwivedi & Bresson (2020) and Vignac et al. (2023) to implement our denoising network $f_\theta$. First, we extract the graph features of $G^t = (\mathbf{X}^t, \mathbf{E}^t)$ following Vignac et al. (2023), resulting in node level feature $\mathbf{F_x} \in \mathbb{R}^{n \times d_x}$, edge level feature $\mathbf{F_e} \in \mathbb{R}^{n \times n \times d_e}$ and graph level feature $y \in \mathbb{R}^{d_y}$. For clarity of presentation, we omit the timestep superscript $t$ for $\mathbf{F_x}, \mathbf{F_e}, y$. These features encode (1) the node and edge types and (2) the graph structure features of $G^t$. We refer the reader to Vignac et al. (2023) for details on the structure features. Global timestep $t$ is encoded in graph level feature $y$. Node level $l_i$ and node-specific local timestep $\tau_i^t$ is encoded in node level feature $\mathbf{F_x}$.

We then incorporate the condition information into graph features.

**Truth table condition as node features**  Each column in truth tables are a series of $\{0, 1\}$ values of one input gate or output gate. The signal values for AND gates are unknown in the condition and therefore we first pad 0 for the values for AND gate. For AIGs with 8 input gates, the padded truth table could be represented as $\mathbf{T} \in \{0, 1\}^{n \times 2^8}$. To compress this representation, we convert each 8-bit sequence in the last dimension of $\mathbf{T}$ to its corresponding integer value. These integer values are then normalized by dividing by 256, resulting in a compressed representation with values in the range [0, 1]. We concatenate the compressed truth table $\mathbf{T}$ with node level features $\mathbf{F_x}$. With a slight abuse of notation, we continue to denote these augmented node features as $\mathbf{F_x}$.

**Property condition as graph level features**  We concatenate the molecule property condition with the graph level feature $y$. With a slight abuse of notation, we continue to denote the resulting graph level feature as $y$.

**Model architecture**  After the condition information is incorporated into graph features $(\mathbf{F_x}, \mathbf{F_e}, y)$, we process the them through several graph transformer layers to update the features. The graph transformer layer is implemented as:

$$\mathbf{Q}, \mathbf{K}, \mathbf{V} = \mathrm{Linear}(\mathbf{F_x}), \mathrm{Linear}(\mathbf{F_x}), \mathrm{Linear}(\mathbf{F_x}) \tag{16}$$

$$s_E, b_E = \mathrm{Linear}(\mathbf{F_e}), \mathrm{Linear}(\mathbf{F_e}) \tag{17}$$

$$s_y^{\mathbf{X}}, b_y^{\mathbf{X}} = \mathrm{Linear}(\mathbf{y}), \mathrm{Linear}(\mathbf{y}) \tag{18}$$

$$s_y^{\mathbf{E}}, b_y^{\mathbf{E}} = \mathrm{Linear}(\mathbf{y}), \mathrm{Linear}(\mathbf{y}) \tag{19}$$

$$\mathbf{atten} = (s_E + 1)\frac{\mathbf{Q}\mathbf{K}^T}{\sqrt{d}} + b_E \tag{20}$$

$$\mathbf{F_x}' = \mathrm{Linear}((s_y^{\mathbf{X}} + 1) \cdot \mathrm{Softmax}(\mathbf{atten})\mathbf{V} + b_y^{\mathbf{X}}) \tag{21}$$

$$\mathbf{F_e}' = \mathrm{Linear}((s_y^{\mathbf{E}} + 1) \cdot \mathbf{atten}) + b_y^{\mathbf{E}}) \tag{22}$$

$$y' = \mathrm{Linear}(\mathrm{Linear}(\mathrm{flatten}(\mathbf{F_x})) + \mathrm{Linear}(\mathrm{flatten}(\mathbf{F_e})) + \mathrm{Linear}(y)) \tag{23}$$

where $d$ is the size of the last dimension of $\mathbf{Q}, \mathbf{K}, \mathbf{V}$ and $\mathbf{F_x}', \mathbf{F_e}', y'$ are updated node, edge and graph level features. Layers are connected by layer normalization and residue operation. The output node and edge features at the last layer are passed through linear layers to predict the probability distribution of node and edge types in the clean graph $G$.

## B.2  Training SeaDAG

The training algorithm for SeaDAG is detailed in Algorithm 1. Note that for AIG generation, the truth table implicitly specifies the numbers of input and output gates, thereby partially determining the node types. For example, we can designate the first $N_{\mathrm{input}}$ nodes as input gates and the last $N_{\mathrm{output}}$ nodes as output gates, with the remaining nodes naturally serving as AND gates. $N_{\mathrm{input}}$ and $N_{\mathrm{output}}$ are the numbers of input and output gates that can be inferred from the truth table. Given this predetermined information, the diffusion of node types becomes unnecessary for AIG graphs. Instead, the model only needs to generate the edge connections. To implement this, we omit the

addition of noise to node types during the forward process and maintain fixed node types in the backward process. Consequently, we exclude the node cross-entropy loss from the model's loss function.

### B.3 Property Decoder for Condition Loss

In this section, we introduce the implementation of property decoder $\phi$. We denote the probability distribution of node types and edge types predicted by the network $f_\theta$ as $p_\theta(\mathbf{X}) \in \mathbb{R}^{n \times k_x}, p_\theta(\mathbf{E}) \in \mathbb{R}^{n \times k_e}$.

**AIG function decoder**  As discussed in the previous section, the node types are fixed in AIG generation and we only need $\mathbf{E}$ to decode the AIG structure. Given the graph structure of an AIG, the logic function represented by an AIG can be directly determined and the output signals are readily computable. However, to ensure that this operation is differentiable and can be incorporated into the training process, several aspects require special consideration:

- Input selection. To determine the inputs (children) for each gate, we employ a softmax operation across all candidate input gates. Candidate gates are those with levels lower than the current gate. For AND gates, we select two inputs, while for output gates, we select one input.
- Edge type selection. The type of each edge (normal or NOT) is determined by computing an edge type score. This score is calculated as $\text{Tanh}(p_\theta(\mathbf{E})_{ij1} - p_\theta(\mathbf{E})_{ij2})$, where $p_\theta(\mathbf{E})$ represents the predicted edge type distribution from nodes $i$ to $j$. A positive score indicates a higher likelihood of a normal edge, while a negative score suggests a higher probability of a NOT edge.

Utilizing the aforementioned differentiable operations, we can compute the output signals as continuous values. The decoded output signals are then compared against the ground truth output signals specified in the condition truth table to compute the condition loss using BCE. Note that the loss computation only involves the output signal portion, as the input signals, which are all possible combinations of input values, are the same for every AIG.

**Molecule property decoder**  Since molecular properties cannot be directly decoded from the molecular structure, we train a separate property prediction model $\phi$ for each of the five condition properties. The architecture of these models is identical to that of the denoising network $f_\theta$, with two difference:(1) $\phi$ takes clean graphs as input rather than the noisy graphs used by $f_\theta$. (2) $\phi$ outputs the predicted molecular property. During the training process, we sample a discrete graph from the predicted distributions $p_\theta(\mathbf{X})$ and $p_\theta(\mathbf{E})$ using the Gumbel-Softmax technique, which allows us to maintain differentiability while working with discrete graph structures. The sampled graph serves as the input to $\phi$ and the predicted value is compared against the property condition using MSE loss.

## C Proofs

**Proof of Equation 10**  During inference, we use the predicted distribution to sample $G^{t-1}$ from $G^t$:

$$p_\theta(G^{t-1}|G^t) = \prod_{1 \le i \le n} p_\theta(x_i^{\tau_i^{t-1}}|G^t) \prod_{1 \le i,j \le n} p_\theta(e_{ij}^{\tau_{ij}^{t-1}}|G^t). \tag{24}$$

To compute $p_\theta(x_i^{\tau_i^{t-1}}|G^t)$, we marginalize over the network predictions:

$$p_\theta(x_i^{\tau_i^{t-1}}|G^t) = \sum_{k \in \{1...k_x\}} p_\theta(x_i^{\tau_i^{t-1}}|x_i = \mathbf{e}_k, G^t) p_\theta(x_i = \mathbf{e}_k|G^t), \tag{25}$$

where $p_\theta(x_i = e_k|G^t)$ is the network prediction and $p_\theta(x_i^{\tau_i^{t-1}}|x_i = e_k, G^t)$ is computed as follows:

$$p_\theta(x_i^{\tau_i^{t-1}}|x_i, G^t) = p_\theta(x_i^{\tau_i^{t-1}}|x_i, x_i^{\tau_i^t}) = \frac{p(x_i^{\tau_i^t}|x_i^{\tau_i^{t-1}}, x_i)q(x_i^{\tau_i^{t-1}}|x_i)}{q(x_i^{\tau_i^t}|x_i)}. \tag{26}$$

$q(x_i^{\tau_i^t}|x_i)$ and $q(x_i^{\tau_i^{t-1}}|x_i)$ could be computed from Equation 4. $p(x_i^{\tau_i^t}|x_i^{\tau_i^{t-1}}, x_i)$ is computed as follows:

$$p(x_i^{\tau_i^t}|x_i^{\tau_i^{t-1}}, x_i) = p(x_i^{\tau_i^t}|x_i^{\tau_i^{t-1}}) \tag{27}$$

$$= \sum_{x_i^{\tau_i^{t-1}+1}} p(x_i^{\tau_i^t}|x_i^{\tau_i^{t-1}+1}, x_i^{\tau_i^{t-1}})q(x_i^{\tau_i^{t-1}+1}|x_i^{\tau_i^{t-1}}) \tag{28}$$

$$= \sum_{x_i^{\tau_i^{t-1}+1}} p(x_i^{\tau_i^t}|x_i^{\tau_i^{t-1}+1})q(x_i^{\tau_i^{t-1}+1}|x_i^{\tau_i^{t-1}}) \tag{29}$$

$$= \sum_{x_i^{\tau_i^{t-1}+1}} \sum_{x_i^{\tau_i^{t-1}+2}} p(x_i^{\tau_i^t}|x_i^{\tau_i^{t-1}+2})q(x_i^{\tau_i^{t-1}+2}|x_i^{\tau_i^{t-1}+1})q(x_i^{\tau_i^{t-1}+1}|x_i^{\tau_i^{t-1}}) \tag{30}$$

$$= \sum_{x_i^{\tau_i^{t-1}+1}} \cdots \sum_{x_i^{\tau_i^{t-2}}} \sum_{x_i^{\tau_i^{t-1}}} \prod_{\tau=\tau_i^{t-1}+1}^{\tau_i^t} q(x_i^\tau|x_i^{\tau-1}). \tag{31}$$

$q(x_i^\tau|x_i^{\tau-1})$ could be computed from Equation 3. $p_\theta(e_{ij}^{\tau_{ij}^{t-1}}|G^t)$ could be calculated likewise.

**Proof of model equivariance** In this section, we prove that the network we use is equivariant to node permutations. For any node permutation $\sigma : [n] \to [n]$, $(\sigma \star \mathbf{X})_{\sigma(i),...\sigma(n)} = \mathbf{X}_{i,...,n}$ and $(\sigma \star \mathbf{E})_{\sigma(i)\sigma(j)k} = \mathbf{E}_{ijk}$. Then for graph $G^t = (\mathbf{X}^t, \mathbf{E}^t)$, we can prove the following:

- Equivariant feature extraction: the graph features are either permutation equivariant (node level and edge level features) or permutation invariant (graph level features). If the features of $G^t$ are $(\mathbf{F_x}, \mathbf{F_e}, y)$, the features for $\sigma \star G^t$ would be $(\sigma \star \mathbf{F_x}, \sigma \star \mathbf{F_e}, y)$.

- Equivariant condition feature: we can easily prove that the truth table condition for AIG is permutation equivariant, i.e. the truth table for $\sigma \star G^t$ being $\sigma \star \mathbf{T}$, and the molecule property condition is permutation invariant.

- Equivariant model architecture: the operations used in the network, i.e. Linear function, self-attention, layer normalization and residue operation, are all permutation equivariant.

Therefore, the entire model is equivariant to node permutations: $f_\theta(\sigma \star G^t, \sigma \star cond) = \sigma \star f_\theta(G^t, cond)$.

**Proof of loss invariance** We employ graph CrossEntropy loss $\mathcal{L}_{graph}$ and condition loss $\mathcal{L}_{cond}$ during training. Based on the equivariance of $f_\theta$ proved above, we can prove $\mathcal{L}_{graph}$ is invariant to permutation.

$$\mathcal{L}_{graph}(\sigma \star G, f_\theta(\sigma \star G^t, \sigma \star cond)) = \sum_{1 \le i \le n} \text{Cross-Entropy}((\sigma \star \mathbf{X})_i, (\sigma \star p_\theta(\mathbf{X}))_i)$$

$$+ \sum_{1 \le i,j \le n} \text{Cross-Entropy}((\sigma \star \mathbf{E})_{ij}, (\sigma \star p_\theta(\mathbf{E}))_{ij}) \tag{32}$$

$$= \sum_{1 \le i \le n} \text{Cross-Entropy}(\mathbf{X}_i, p_\theta(\mathbf{X})_i)$$

$$+ \sum_{1 \le i,j \le n} \text{Cross-Entropy}(\mathbf{E}_{ij}, p_\theta(\mathbf{E})_{ij}) \tag{33}$$

$$= \mathcal{L}_{graph}(G, f_\theta(G^t, cond)) \tag{34}$$

For the truth table decoder for AIG: $\phi(G) = \mathbf{T}$, the operation is equivariant: $\phi(\sigma \star G) = \sigma \star \mathbf{T} = \sigma \star \phi(G)$. Therefore, we can prove $\mathcal{L}_{cond}$ for AIG generation is permutation invariant:

$$\mathcal{L}_{cond}(\sigma \star cond, \sigma \star \hat{G}) = \text{BCE}(\sigma \star cond, \phi(\sigma \star \hat{G})) \tag{35}$$

$$= \text{BCE}(\sigma \star cond, \sigma \star \phi(\hat{G})) \tag{36}$$

$$= \text{BCE}(cond, \phi(\hat{G})) \qquad \leftarrow \text{ Invariant BCE loss} \tag{37}$$

$$= \mathcal{L}_{cond}(cond, \hat{G}) \tag{38}$$

For the molecule property decoder, $\phi$ has the same equivariant model structure as $f_\theta$ and is permutation invariant: $\phi(\sigma \star G) = \phi(G)$. Similarly, we can prove its invariance:

$$\mathcal{L}_{cond}(cond, \sigma \star \hat{G}) = \text{MSE}(cond, \phi(\sigma \star \hat{G})) \tag{39}$$

$$= \text{MSE}(cond, \phi(\hat{G})) \qquad \leftarrow \text{ Invariant MSE loss} \tag{40}$$

$$= \mathcal{L}_{cond}(cond, \hat{G}) \tag{41}$$

In conclusion, the total training loss is invariant to node permutation.

## D  SAMPLING FROM SEADAG

### D.1  SAMPLING ALGORITHM

To sample a DAG from SeaDAG, we first sample the number of levels in the DAG: $N \sim p(N)$, where $p(N)$ is the distribution of number of levels observed in the training set. Then we sample the size, i.e. the number of nodes, for each level: $M_i \sim p(M_i|i)$, for $i = 0, 1, ..., N-1$, where $i$ is the level index, $M_i$ is the level size and $p(M_i|i)$ is the level size distribution of level $i$ in the training set. However, for AIG generation, the sizes of the first and last levels are predetermined by the truth table condition. The size of the first level is set to $N_{\text{input}}$ which is the number of input gates and the size of the last level is set to $N_{\text{output}}$ which is the number of output gates. In the case of molecule generation, the size of the last level is set to one because of the tree structure of junction trees. Consequently, the sizes of these particular levels are not sampled but are instead predefined.

Once the number of levels and their respective sizes are determined, we can determine the level assignment $l_i$ for each node $i$. Subsequently, we sample an initial random graph based on the marginal distributions of node types $\mathbf{m_X}$ and edge types $\mathbf{m_E}$ and use the denoising model $f_\theta$ to gradually remove noise from the random graph. Algorithm 2 illustrates the above process.

### D.2  AIG GENERATION SPECIFICATION

We now detail the procedure for parsing an AIG from the generated DAG $G = (\mathbf{X}, \mathbf{E})$ obtained from Algorithm 2. As previously discussed, the node types are predetermined during AIG generation, so we only need to parse the gate connections. For each AND gate $i$, we sample its two inputs from its children set $\text{Ch}(n_i)$, which is defined by the $i$-th column of $\mathbf{E}$. The edge from a child gate $j$ to gate $i$ is a normal edge if $\text{argmax}(\mathbf{E}_{ji}) = 1$ or a negation edge if $\text{argmax}(\mathbf{E}_{ji}) = 2$. If $|\text{Ch}(n_i)| < 2$, we introduce a new input gate to the AIG that constantly outputs a signal of 0, and connect this new gate to gate $i$. For output gates, we sample one input for each in a similar manner. Once the inputs have been determined for each AND gate and output gate, we remove floating gates from the AIG, which are defined as the AND gates and input gates that are not in the cone of any output gates. Finally, we simulate the resulting AIG to obtain its output for evaluation. For all baselines and our model, we generate 10 AIGs for a truth table and take the one with highest function accuracy.

### D.3  MOLECULE GENERATION SPECIFICATION

To parse a molecule junction tree from the generated DAG $G = (\mathbf{X}, \mathbf{E})$, we first determine the node type for every node by computing $\text{argmax}(\mathbf{X}_i)$. Next, starting from the nodes at lower levels, we sequentially determine the children of each node $i$ by selecting the nodes in $\text{Ch}(n_i)$ that have not yet been assigned a parent. We then designate the node without a parent as the root of the junction tree. In cases where multiple nodes lack a parent, we randomly assign a parent from nodes at higher

---

**Algorithm 2** SeaDAG sampling algorithm

---

**Input:** Condition $cond$, Denoising model $f_\theta$, function $\mathcal{T}$ to map global timestep $t$ and node level $l_i$
 to local timestep
**Output:** Generated graph $G = (\mathbf{X}, \mathbf{E})$
 Sample number of levels $N \sim p(N)$
 Sample size of each level $M_i \sim p(M_i|i)$, for $i = 0, 1, ..., N-1$
 $n \leftarrow \sum_{i=0}^{N-1} M_i$
 Sample $G^T \sim \prod_{1 \le i \le n} \mathbf{m_X} \times \prod_{1 \le i,j \le n} \mathbf{m_E}$
 **for** $t$ from $T$ to 1 **do**
  Compute node-specific local timestep $\tau_i^t \leftarrow \mathcal{T}(t, l_i)$
  Compute edge-specific local timestep $\tau_{ij}^t \leftarrow \mathcal{T}(t, l_j)$
  $p_\theta(\mathbf{X}), p_\theta(\mathbf{E}) \leftarrow f_\theta(G^t, cond)$
  $p_\theta(x_i^{\tau_i^{t-1}}|G^t) \leftarrow \sum_{k \in \{1...k_x\}} p_\theta(x_i^{\tau_i^{t-1}}|x_i = \mathbf{e}_k, G^t) p_\theta(x_i = \mathbf{e}_k|G^t)$
  $p_\theta(e_{ij}^{\tau_{ij}^{t-1}}|G^t) \leftarrow \sum_{k \in \{1...k_e\}} p_\theta(e_{ij}^{\tau_{ij}^{t-1}}|e_{ij} = \mathbf{e}_k, G^t) p_\theta(e_{ij} = \mathbf{e}_k|G^t)$
  Sample $G^{t-1} \sim \prod_{1 \le i \le n} p_\theta(x_i^{\tau_i^{t-1}}|G^t) \times \prod_{1 \le i,j \le n} p_\theta(e_{ij}^{\tau_{ij}^{t-1}}|G^t)$
 **end for**
 **return** $G^0$

---

levels for all such nodes except the one at the highest level, which becomes the root of the tree. Once we obtain the junction trees, we follow the methodology outlined by Jin et al. (2018) to assemble these trees into complete molecules for subsequent evaluation.

# E  EXPERIMENT SETTINGS

## E.1  DATASET STATISTICS

Table 8: Dataset information.

| Dataset | #Training graphs | #Valid. graphs | #Test graphs | Average #nodes | Node types | Edge types |
|---------|------------------|----------------|--------------|----------------|------------|------------|
| AIG     | 12950            | 1850           | 3700         | 21.00          | 3          | 3          |
| QM9     | 96663            | 19659          | 10000        | 5.36           | 1839       | 2          |
| ZINC    | 218969           | 24333          | 4973         | 14.38          | 780        | 2          |

We list the statistics of the datasets in Table 8.

## E.2  TRAINING PARAMETERS FOR SEADAG

For AIG generation, we choose $T = 500, \lambda = 1., \beta = 32$ and use 8 graph transformer layers in the network $f_\theta$. We train the model for 1000 epochs with learning rate 0.0002 and batch size 256. We use AdamW optimization (Loshchilov & Hutter, 2019) with weight decay coefficient $1.0e - 12$. For molecule generation on QM9 (ZINC), we choose $T = 500, \lambda = 1., \beta = 32$ and use 8 graph transformer layers in the network $f_\theta$. We train the model for 1000 epochs with learning rate 0.0002 and batch size 300 (320). We use AdamW optimization with weight decay coefficient $1.0e - 12$. We trained 7 molecule property prediction models in total (5 for QM9 and 2 for ZINC). We use 3 graph transformer layers in their networks. The remaining training configurations are largely consistent with those used for the denoising network.

## E.3  BASELINES

**AIG generation**  We extend our baselines to achieve conditional AIG generation in the following way:

- SwinGNN: we incorporate truth table condition into graph features by mapping a truth table to a vector embedding using Linear layers and concatenating the embedding with graph level features. We choose this strategy based on empirical results. We train the model with the following parameters: num_steps = 256, model_depths = [1, 3, 1], patch_size = 4, window_size = 4, batch_size = 512, learning_rate = 0.0001, epoch = 1000, weight_decay = 0. Since SwinGNN is a continuous diffusion model, the generated continuous values need to be discretized. We choose a threshold that leads to the highest function accuracy.

- Pard: to incorporate the condition, we treat the truth table as node-level features as in our model. Since Pard is a block-wise autoregressive diffusion model, the output gates are not generated until the last steps and their signals can not be seen by nodes generated before. To make the output signals available during the entire generation process, we flatten the output signals and concatenate it with every node feature. Different from the original work where the graphs are generated block-by-block and the blocks are divided based on node degrees, we modify Pard to generate DAGs level-by-level like our model. Following their proposed procedure, we train two models: a model to predict the size of the next level to generate and a model that performs diffusion generation for the next level with predicted size. The level size prediction model is trained with the following parameters: hidden_size = 256, num_layers = 8, batch_size = 256, dropout = 0., epochs = 500, learning_rate = 0.0002 with decay = 0.5 and weight decay coefficient = 0.01. The local diffusion model is trained with the following parameters: cross-entropy loss coefficient = 0.1, diffusion steps = 100, hidden_size = 256, num_layers = 8, batch_size = 256, dropout = 0., epochs = 500, learning_rate = 0.0002 with decay = 0.5 and weight decay coefficient = 0.01. We use cosine learning rate scheduler and AdamW optimization for both models.

- DiGress: we incorporate the truth table condition in the same manner as in our SeaDAG model, namely by compressing the truth table and concatenating it with node level features. Then we train the model using the following parameters: transition = marginal, diffusion_steps = 500, num_layers = 8, $\lambda = 1$, epochs = 1000, batch_size = 640, learning_rate = 0.0002 and weight decay coefficient = 1.0e-12. We use AdamW optimization.

**Molecule generation** For baselines employed in molecule generation, we follow the methods outlined in the original papers to perform both conditional and unconditional generation. Specifically, for conditional generation using DiGress, we employed the discrete regressor guidance strategy introduced in their work. We first train an unconditional generation model and then train five property regressors for the five property conditions in our settings. These regressors were then utilized to guide conditional generation. For more details on the baselines, we direct the reader to the original works.

### E.4 CONDITIONAL GENERATION METRICS

We report two metrics when evaluating the conditional AIG generation performance: (1) Validity: Since an AIG is valid as long as the gates receive the correct number of inputs (two for AND gates and one for output gates), we compute the percentage of AND and output gates that have correct number of inputs as the validity of generated AIGs. (2) Accuracy: As discussed in Section B.3, only output signals participate in the computation of loss function and accuracy evaluation since the input signals are always fixed. Therefore, we take the output signals from condition truth tables and those of the generated AIGs and compute the element-wise accuracy.

We report the mean absolute error between the condition properties and the oracle-predicted properties of molecules. To calculate the predicted properties of generated molecules, we closely follow the procedure used in You et al. (2023).

### E.5 GRAPH QUALITY EVALUATION METRICS

To evaluate the quality of generated molecules, we employ the metrics in the benchmark proposed by Polykovskiy et al. (2020). Metrics including Validity, FCD, Unique and Novelty are computed using the provided evaluation codebase. The metric NSPDK are computed using the code provided by Gao et al. (2024).

### E.6 MOLECULE OPTIMIZATION VIA CONDITIONAL GENERATION

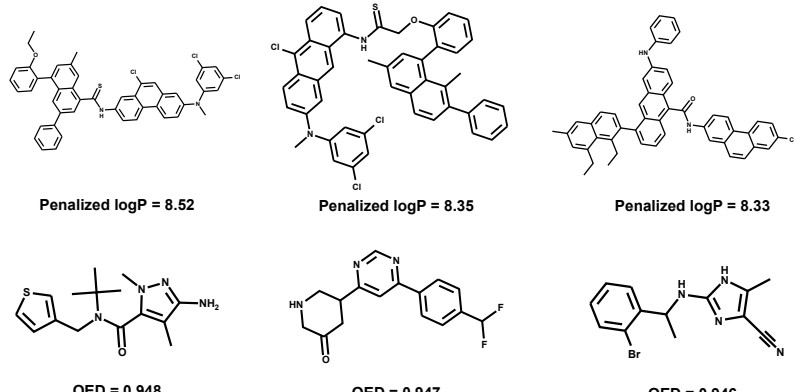

Figure 8: The top 3 molecules for penalized logP and QED found by SeaDAG.

We demonstrate that we could leverage the conditional generation provided by SeaDAG to generate molecules with optimized properties. We first train SeaDAG on ZINC dataset to generate molecules conditioned on the penalized logP and QED. Then we sample 2000 molecules for each property and take the top 3 for evaluation. For penalized logP, we sample molecules with condition values ranging from 7 to 10. For QED, we sample molecules with condition values ranging from 0.94 to 0.99. Figure 8 presents the top 3 molecules we found for the two properties.

## F MORE SAMPLED GRAPHS

We provide more DAGs sampled from our SeaDAG in Figure 9 and Figure 10.

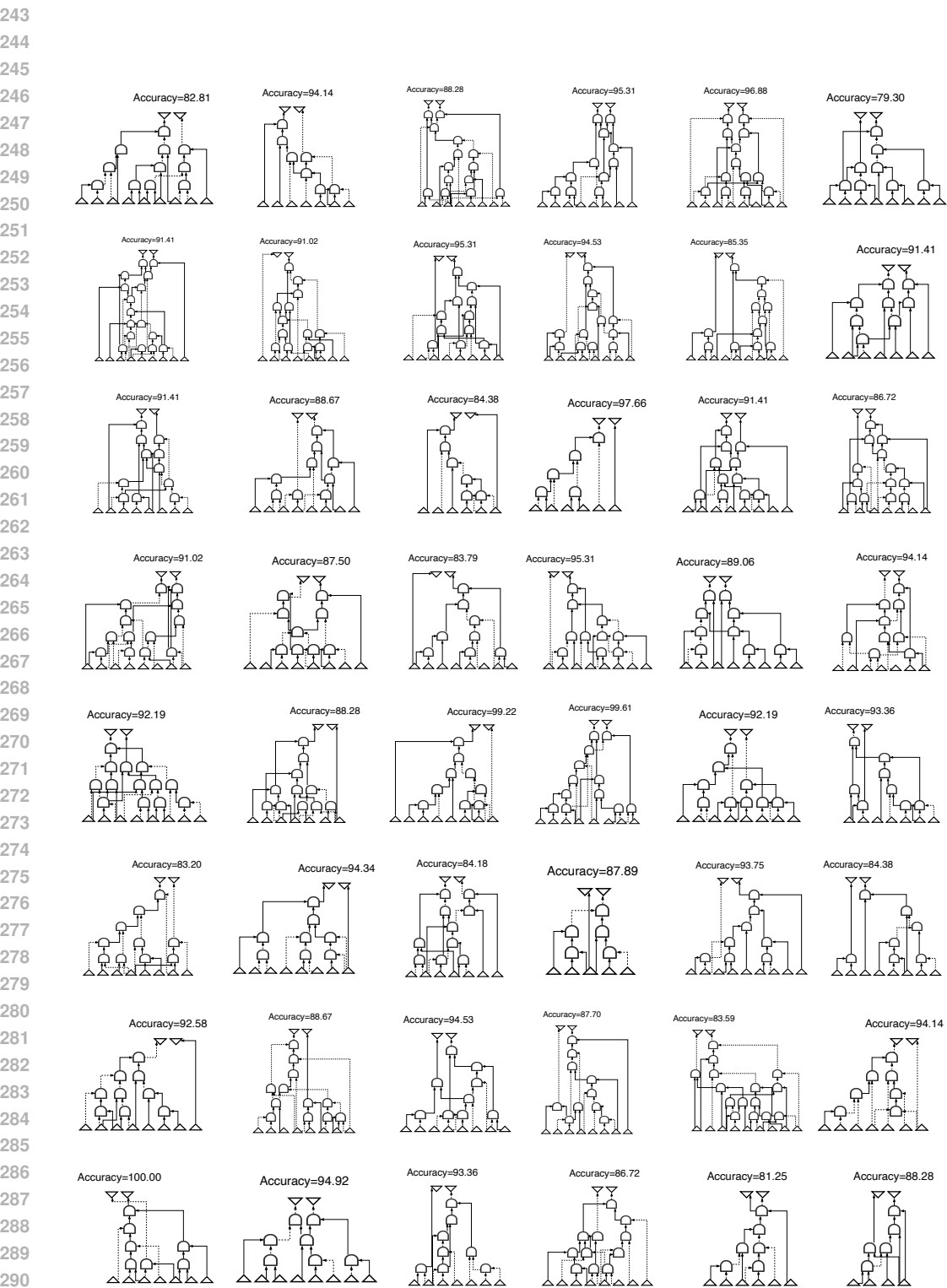

Figure 9: AIGs sampled from SeaDAG.

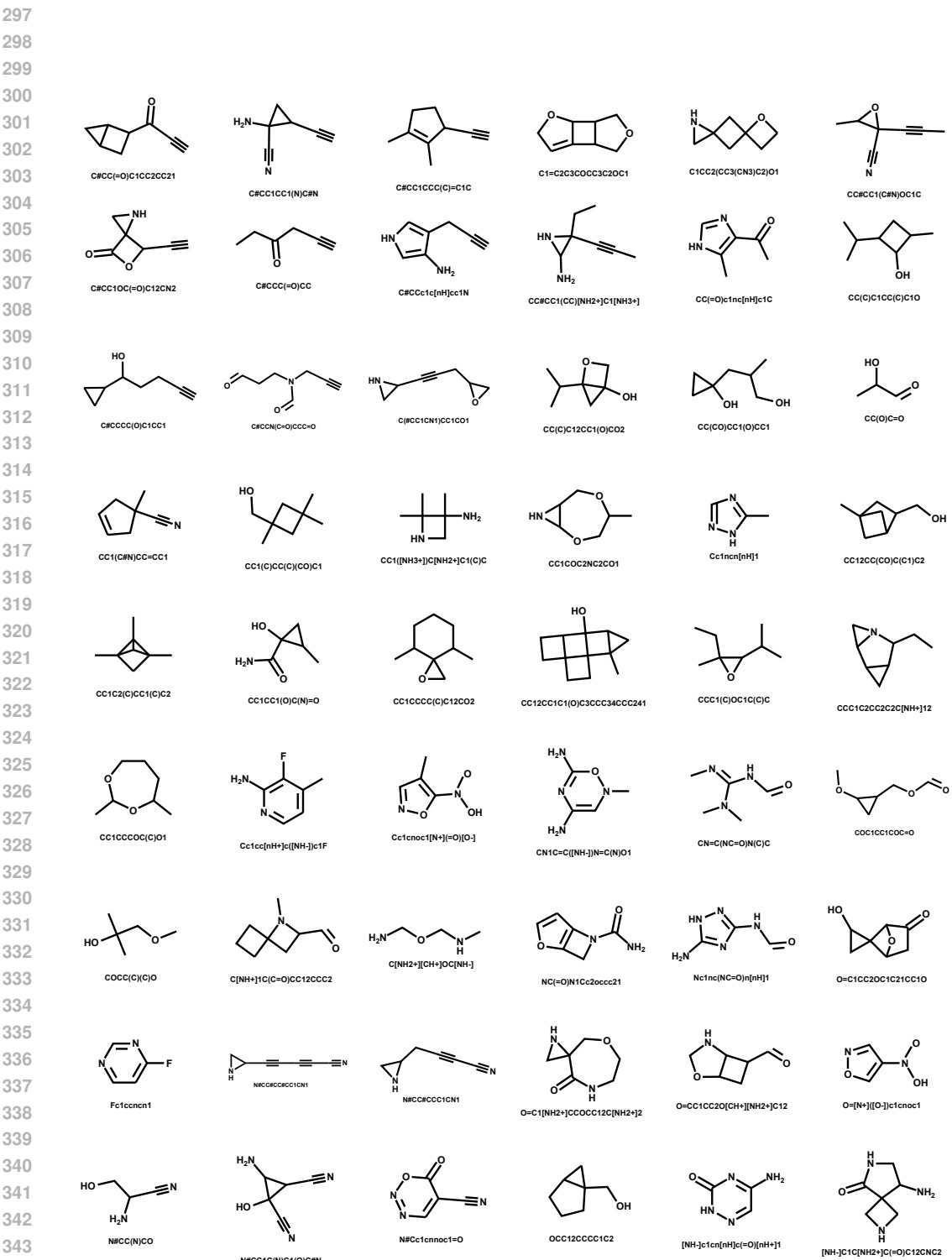

Figure 10: Molecules sampled from SeaDAG.

