# OpenReview forum: "SeaDAG: Semi-autoregressive Diffusion for Conditional Directed Acyclic Graph Generation"
_ICLR.cc/2025/Conference — ICLR 2025 Conference Withdrawn Submission_

### Official Review · Reviewer_iPZe · 2024-10-31

**Soundness:** 2
**Presentation:** 2
**Contribution:** 2
**Rating:** 3
**Confidence:** 3

**Summary:**

The paper proposes SeaDAG, a generative model for directed acyclic graphs (DAGs). SeaDAG specializes in graph diffusion to DAGs by varying the noise schedule across layers of the DAG, thereby denoising top (or bottom) layers faster than bottom (or top) layers. In addition, the authors present a scheme for training SeaDAG conditionally: SeaDAG is explicitly conditioned on some properties during training and the authors introduce a novel loss term, called "condition loss". Here, a differentiable decoder attempts to infer the property of the predicted clean graph, and deviations from the provided conditioning are penalized. The method is evaluated in the tasks of generating And-Inverter Graphs and small molecules (via junction trees).

**Strengths:**

- The idea of denoising different levels of a DAG at different speeds is reasonable and intuitive. It appears that this substantially improves the validity of generated graphs.
- Molecule generation is currently one of the few graph-generation tasks that are close to real-world applications. Therefore, I greatly appreciate the results for this task and I believe that they could be emphasized even further.

**Weaknesses:**

- The scope of this paper is limited to the generation of DAGs. While DAGs are important in many domains and may warrant special treatment, the impact may be somewhat limited in comparison to general-purpose graph generative models.
- Given that this work focuses exclusively on generating DAGs, I would expect an evaluation of baselines that also specifically address this task, e.g. LayerDAG (Li et al., 2024a). This seems especially important since line 45ff appears to frame SeaDAG as an improvement over LayerDAG. This point applies both to experiments and related work.
- For a conditional generation with DiGress, the authors use the guidance approach. This is a post-hoc approach to injecting conditions into a trained unconditional model. For SeaDAG, however, the authors perform conditional training. To ensure a more fair comparison, I would suggest a comparison to a DiGress model that is trained conditionally using the most commonly used approach [1]. If such a comparison is not feasible, it's important to explain why this approach was not used and discuss any limitations of the current comparison. It is worth noting that this approach was also mentioned as a possible extension in the DiGress paper (even though computationally more expensive). Therefore, the point (3) raised in line 53 is debatable.
- JT-VAE is closely related to SeaDAG's approach to molecule generation. I was surprised that it was not included as a comparison partner in Tables 2 and 3. A conditional JT-VAE would also be a highly relevant baseline. If conditioning JT-VAE was not possible, a comparison of JT-VAE and an unconditional SeaDAG model would be crucial for understanding SeaDAG's advantages.
- On the molecule generation task (e.g. Table 3), the compared graph generative models (e.g. DiGress) generate the whole molecular graphs, while SeaDAG generates junction trees. An important baseline would be to use DiGress to generate junction trees directly. It is not clear to me whether the improvements stem from the proposed model or simply from the reformulation of the problem as junction tree generation.
- The AIG-generation task is not convincing. From the paper, It is not clear how challenging this task is and I believe more background should be provided for readers unfamiliar with the area of logic synthesis. In addition, I think benchmarking the proposed method on some existing datasets, e.g. those used by LayerDAG (Li et al., 2024a), would be more convincing.
- The condition loss seems to be a highly heuristic modification. As noted in A3, the empirical evidence for its usefulness remains mostly in the AIG task. It would be useful if the authors provided some theoretical justification for this choice, such as analyzing how the condition loss affects the optimization landscape or relating it to existing theoretical frameworks for conditional generation.
- The paper would benefit from a comprehensive ablation study in the main text. While the authors propose two distinct techniques, namely semi-autoregressive generation and condition loss, they only present empirical results for these methods in combination. Separating and analyzing the individual contributions of each technique would provide clearer insights into their respective impacts on performance. A possible ablation study design could be evaluating models with 1) only semi-autoregressive generation, 2) only condition loss, 3) both techniques, and 4) neither technique (baseline), on a key benchmark task.
- The absence of unconditional generation results represents a missed opportunity. Testing on unconditional tasks, particularly for molecule generation, would have allowed for two key improvements. First, it would have enabled evaluating the semi-autoregressive approach independently from the condition loss. Second, and more significantly, it would have facilitated a more comprehensive evaluation by: 1. Enabling fairer comparisons with existing QM9 literature; 2. Allowing testing on more sophisticated datasets like MOSES and Guacamol, which contain more complex molecular structures than QM9's limit of 9 heavy atoms. This expanded evaluation scope would have provided more compelling evidence for the method's practical applicability.

#### Minor points:
- Line 44-70: This paragraph appears to be an important motivation but is not yet clear to me. In (1), it is unclear what "information flow" refers to, and the statement that this is an issue appears to lack evidence (LayerDAG is not evaluated in experiments).
- The notation on Line 106 suffers from symbol overloading: the variable $n$ is used both to represent the cardinality of vocabulary set $\mathcal{V}$ and as an index variable. This creates confusion, particularly evident in the expression $n_n$ which is mathematically ambiguous. The notation should be revised to use distinct symbols for these different purposes.
- Lines 107 and 108: I suggest avoiding starting a sentence with a mathematical symbol.
- Line 179: I would consider simplifying notation, as there are both $e_i$ (referring to an edge type) and bold $\boldsymbol{e}_i$ (referring to a standard basis vector).
- Eq (3) and (4): I think that one would want to do a right multiplication by $Q^t$, instead of a left multiplication.
- Line 303: GraphARM is cited in the context of conditional guidance. Could the authors elaborate on how GraphARM uses a guidance approach?
- Line 374: It is unclear what the definition of "structural validity" exactly refers to.

[1] Emiel Hoogeboom, Vicctor Garcia Satorras, Clement Vignac, and Max Welling. Equivariant diffusion for molecule generation in 3D. In International Conference on Machine Learning, pp. 8867–8887. PMLR, 2022. 6, 7, 20.

**Questions:**

1. How does unconditional SeaDAG perform on QM9, Guacamol, and MOSES?
2. How does SeaDAG perform in comparison to JT-VAE (with or without conditioning)?
3. How do general-purpose graph generators (e.g. DiGress) perform on QM9 when trained to generate junction trees instead of full graphs?
4. Does SeaDAG encode the layer index into node attributes that are supplied to the model?
5. Is SeaDAG constrained to only produce acyclic graphs? I.e. do you constrain the model to only produce arcs between neighboring levels?

---

### Official Review · Reviewer_BEVw · 2024-10-31

**Soundness:** 2
**Presentation:** 4
**Contribution:** 2
**Rating:** 5
**Confidence:** 4

**Summary:**

This paper proposes SeaDAG, a method for generating directed acyclic graphs (DAGs) using diffusion. At its core, SeaDAG performs diffusion based on the discrete graph-diffusion method DiGress (Vignac, et. al, 2022). In contrast with DiGress, however, SeaDAG performs diffusion differently in two ways.

Firstly, because DAGs have an implicit topological ordering (i.e. every node can be assigned a “level” or “depth”), SeaDAG sets the noise schedule differently for nodes and edges at different levels. For example, nodes with a low level can be given a slow noise schedule, so that in reverse diffusion, it is defined/fixed more quickly than nodes at a high level. Secondly, SeaDAG performs conditional diffusion by training the diffusion model with a conditioning signal in the loss. In particular, during training, a noisy graph is sampled and the diffusion model is trained to reconstruct the clean (denoised) graph. In order to provide a conditioning signal for a property of interest, another model (e.g. a classifier) predicts the property of interest from the denoised graph (for each example during training), and penalizes the model if the property does not match the desired conditioning signal.

The authors demonstrate SeaDAG on a synthetic dataset of circuits, and two molecular datasets (where the molecules are represented in junction-tree format to turn them into DAGs): QM9 and ZINC. The authors show that SeaDAG can produce DAGs (unconditioned or conditioned) which are comparable in quality to some other general graph-diffusion methods.

**Strengths:**

### Clear writing and description of what was done

The manuscript is clearly written. The methodological novelties are clearly described and the procedure is largely made clear by the main text and the thorough supplement. The connection with previous works is also well explained. The experimental section presents the empirical results in a cleanly, and the contents of each experiment are well described.

### Compelling results on utility of non-uniform noise schedule on AIGs (Figure 5)

The results on the AIG graphs showing the utility of non-uniform noise (i.e. semi-autoregressive generation) is fairly convincing. Assuming that the same sampling procedure (as described in the supplement) is used in both SeaDAG panels of Figure 5, this is a rather compelling result which shows a specific benefit of the unique noise schedule.

### Good comparisons to other neural networks to many relevant works

In many of the analyses, there are many other works being compared to, which show that SeaDAG achieves comparable generative performance in the tested datasets. Additionally, the performance metrics used are reasonable diverse and thorough.

**Weaknesses:**

### Missing comparison to classifier-free diffusion

One of the two technical contributions of this work is the condition loss, where generating DAGs conditioned on some label is performed using an auxiliary classifier/regressor for that label, during training. One of the major claims that motivates this development is the argument that it is better to build conditioning signals into the training process rather than separating it during inference time (Section 2.3.2).

The most informative and meaningful comparison to demonstrate this claim would be to train SeaDAG on the same datasets (with and without the semi-autoregressive noise schedule), but apply classifier-free diffusion (Ho & Salimans, 2021), where the conditioning signal is fed into the neural network as an input itself (modifying the architecture to feed in the conditioning label embedding in a way that is similar to other classifier-free conditional graph-diffusion works). This method of classifier-free guidance is arguably by far the most commonly used form of conditional diffusion in the literature currently, and it is also a way to build conditional diffusion models by incorporating the conditioning signal at training time.

There doesn’t seem to be any comparison to this classifier-free guidance anywhere, even in Tables 2 – 4, which show results on conditional diffusion. This comparison is crucial to understand the performance of SeaDAG’s condition loss.

### Missing comparison to DiGress or LDM-3DG in Table 4

In addition to missing the comparison with classifier-free guidance, Table 4 (conditional molecular generation) does not compare to DiGress of LDM-3DG. In particular, DiGress is a close analog of SeaDAG because it uses the same diffusion process (minus the semi-autoregressive noise schedule). It would be more meaningful to also show a comparison with these two works on the ZINC dataset.

### Robustness of noise schedule

There aren’t currently results on the robustness of SeaDAG to the noise schedule. Since the noise schedule (which defines the semi-autoregressive diffusion process) is the main contribution of this work, it is important to show the robustness of SeaDAG to the selected noise schedule, as well as some reasoning for why the noise schedule is designed as it is (i.e. equations 5 and 6).

### Somewhat limited ablation studies

The ablation studies in Section A.3 are very appreciated, but it would also be great to include a row in Tables 6 and 7 which shows SeaDAG (same dataset and architecture as the other rows), but trained without the semi-autoregressive noise schedule and without the conditioning loss (and instead use the method employed by DiGress; if this is just equivalent to DiGress, then it would be good to mention that and reproduce that row).

It also seems that from these tables, the conditioning loss seems to actually hurt the quality of the generated objects on average.

### Sampling using SeaDAG starts with more given information

When sampling graphs using SeaDAG, the number of levels in the DAG, as well as the number of nodes in each level, are effectively pre-determined by sampling from the empirical distribution. In comparison, other methods don’t seem to have that information given at inference time. As such, the experiments which compare the quality of DAGs generated by SeaDAG versus other methods are skewed fairly heavily toward SeaDAG’s favor. This makes these experiments quite a bit weaker when trying to show that SeaDAG has comparable or better performance than these other methods.

If there is a way to also inject this same information into other methods, this would be a more fair comparison. As of now, SeaDAG seems to have an unfair advantage because it is given a lot more information about the graph structure at the outset.

### Limited datasets

This is a more minor point, but the datasets are a bit limited in this work. The AIG dataset is interesting, but ultimately fairly simplistic and synthetic. QM9 and ZINC are great real-world datasets, but they are both molecular and each molecule is limited in size. Perhaps there are additional meaningful DAG datasets (with more complexity and interesting conditioning signals) which can be shown, as well.

**Questions:**

### How is the noise schedule chosen?

In particular, how is $\beta$ selected? Additionally, why was a bottom-up noise schedule used for AIGs and a top-down schedule used for molecules? Is there any intuition as to why one is better than the other for different datasets?

### Is the network $\phi$ (for the conditioning loss) pre-trained or trained end-to-end with the diffusion model?

### Does the conditioning loss and/or semi-autoregressive noise schedule lead to more or less efficient training?

Compared to a uniform noise schedule, does SeaDAG take less time to train or sample from? Compared to classifier-free guidance (and DiGress’ conditional diffusion), does SeaDAG’s conditioning loss take more time to train?

---

### Official Review · Reviewer_nvSf · 2024-11-03

**Soundness:** 3
**Presentation:** 2
**Contribution:** 1
**Rating:** 3
**Confidence:** 4

**Summary:**

The paper proposes a semi-autoregressive discrete denoising diffusion model for the conditional generation of directed acyclic graphs (DAGs). The core part of this model is a combination of the discrete denoising diffusion model for general graphs [1] and the multilevel diffusion model for text [2]. The fundamental motivation behind this design choice is to ensure that each layer in a DAG is denoised at a different speed, which may benefit the performance due to a different degree of interactions between the layers. The secondary (optional) part of the model involves a property decoder to improve the model's ability to generate graphs conditionally on desired properties. The experiments demonstrate the conditional generation of (i) logic circuits based on truth tables and (ii) molecules based on chemical properties, comparing the proposed model with several existing benchmarks.

**Strengths:**

The paper is experimentally thorough.

The examples of generating logic circuits are nice and interesting.

The design choices are nicely illustrated throughout the text.

**Weaknesses:**

Major:

The proposed model is a direct combination of [1] and [2]. The authors use [1] as the template for their diffusion model. The main difference to [1] is in the way $p\_\theta(x^{\tau^{t-1}\_i}\_i|x\_i=e\_k,G^t)$ in equation 11 is computed. The authors borrow equation 10 from [2] and the accompanying time step schedules to compute this distribution. Then, they only need to instantiate this equation in the context of Markov transition matrices.

Section 2.1 is nicely written. However, its focus should be somewhere else. The paper is about a semi-autoregressive discrete diffusion model. Therefore, the structure of this section can be, e.g., as follows: (i) discrete diffusion models, (ii) autoregressive diffusion models, and (iii) answering where the semi-autoregressive diffusion model lies between these two cases. The description of domain-specific instances of DAGs should be left in the appendix. In the current form, it is not easy to separate the paper's true methodological contributions and originality in Section 2.2.

Section 3 does not discuss any relation to diffusion models for DAGs. Previous work on multilevel diffusion or semi-(non-)autoregressive models needs to be covered. The authors can look at the related work section in [2] to initiate their search for references.

Minor:

Section I should do a better job articulating the paper's original contributions. For example, it would be better if the authors connect the points (1-3) in the second paragraph with (1-3) in the second last paragraph (though one of these will have to use different points, say, (a-c)).

Please define $p_\theta$ in $f_\theta$ in line 215. If the reader is unfamiliar with [1], it will be hard to understand the redefinition of $f_\theta$ in line 257.

**Questions:**

Please provide more details on the similarities between the proposed model and the one in [3].

Line 215: ``*Specifically, we extend the graph transformer architecture from Dwivedi \& Brensson (2020).*'' How do the extensions compare to those described in [1]?

Line 338: ``*We adopt the standard dataset split, utilizing 100k graphs for training and 20k for validation.*'' Does it mean the experiments use the predefined set of training and validation indices as, e.g., in [4] (https://github.com/calvin-zcx/moflow/blob/master/data/valid\_idx\_qm9.json)?

Line 458: "*Note that for QM9 dataset, a higher Novelty score does not
indicate better performance, but rather suggests a deviation from the dataset's distribution, as QM9
comprehensively enumerates molecules within specific constraints (Vignac \& Frossard, 2022).*" Can this statement be verified? Many methods do achieve all metrics close to 100\%.

Why is there no SeaDAG in the `unconditional' part of Table 3?

Are the molecules in Figure 10 only from a model trained on the QM9 dataset?

Please also show a separate set of samples for the Zinc250k dataset.

Why is there no table with molecular metrics also for the Zinc250k dataset? At least in the supplementary material.

DiGress [2] uses the empirical distribution of the number of nodes in the training graphs to sample the number of nodes in the newly generated graphs. For QM9, the model can sample at most nine atoms. However, some molecules in Figure 10 have 12 atoms. How does the model extrapolate the empirical distribution to more than nine atoms?

We have to pick the root node to convert a molecular graph to a tree graph. Did the authors try different choices for the root node? If yes, did they observe any performance deviations?

Regarding the experiments in Figure 6. Can the authors please show a histogram of the number of input gates computed from the ground truth, DiGress, and SeaDAG samples?

[1] Vignac, C., Krawczuk, I., Siraudin, A., Wang, B., Cevher, V. and Frossard, P., DiGress: Discrete Denoising diffusion for graph generation. In The Eleventh International Conference on Learning Representations.

[2] Wu, T., Fan, Z., Liu, X., Zheng, H.T., Gong, Y., Jiao, J., Li, J., Guo, J., Duan, N. and Chen, W., 2023. Ar-diffusion: Auto-regressive diffusion model for text generation. Advances in Neural Information Processing Systems, 36, pp.39957-39974.

[3] Li, M., Shitole, V., Chien, E., Man, C., Wang, Z., Zhang, Y., Krishna, T. and Li, P., LayerDAG: A Layerwise Autoregressive Diffusion Model of Directed Acyclic Graphs for System. In Machine Learning for Computer Architecture and Systems 2024.

[4] Zang, C. and Wang, F., 2020, August. Moflow: an invertible flow model for generating molecular graphs. In Proceedings of the 26th ACM SIGKDD international conference on knowledge discovery \& data mining (pp. 617-626).

---

### Official Review · Reviewer_Pb1P · 2024-11-09

**Soundness:** 2
**Presentation:** 3
**Contribution:** 2
**Rating:** 6
**Confidence:** 3

**Summary:**

The paper proposes SeaDAG which is a semi-autoregressive diffusion-based DAG generation model. SeaDAG utilizes discrete graph denoising diffusion, which operates on the transition metrices of marginal node and edge types. To leverage the layerwise structure of DAGs, SeaDAG will have different diffusion speed for different layers, and the reconstruction loss of the graph is a combined cross entropy of node and edge losses. To incorporate the conditional graph properties and conditional generation, SeaDAG introduces the graph property decoding module, where the training loss will be combined conditional loss and graph loss. The authors evaluate SeaDAG on various experiments including AIG and molecules.

**Strengths:**

(1) The paper is written with a logical flow and clear exposition, and the authors provide an algorithm pseudocodes that makes the algorithm easy to understand.

(2) The empirical performance of SeaDAG is promising, demonstrating improvements across a range of benchmark tests and scenarios like  AIG and molecules.

(3) The authors have provided additional theoretical analysis to show that SeaDAG is permutation equivariant with the utilized
graph generation and that the proposed loss is invariant to node permutations.

(4) The authors provide a detailed experiment setting for training SeaDAG in AIG and molecule dataset. The training details with SeaDAG and comparison methods make it easier to replicate the experiments.

**Weaknesses:**

(1) For autoregressive between the layers, the authors introduce the different diffusion speed for noise schedules. However, it is unclear just diffusing between different time steps and layers can capture the interdependencies between the layers. It can hardly be called "autoregressive" since the whole graphs are generated instead of components of graphs generating.

(2) There is lack of ablation study about how different choices of hyper-paramter $\beta$ or how different diffusion speed could impact the performances.

(3) While the SeaDAG algorithm is trained on conditional loss terms, the comparison methods in the experiments—such as Pard, DiGress, and SwinGNN—do not explicitly incorporate conditional loss information. These existing baseline methods are graph generative models that do not constrain themselves to generate only DAG data. In fact, many of them tackle a more challenging problem: determining an ordering to generate graphs, which is inherently more complex than generating a DAG where node hierarchy and structure are predefined. Consequently, direct comparisons between SeaDAG and these methods may not be entirely fair.

(4) In addition to the experiment results, the authors should also report the training time and memory sizes of different methods for a fair comparison, for example, training time per epoch, total training time, peak memory usage during training, and model size in terms of parameters.

(5)  During sampling and inference time, the authors mention that the hierarchical structure, as the number of levels and their sizes from distributions, are sampled from the training data. However, there is no adjustment of the number of levels or sizes during the sampling process, which could lead to mode collapsing and losing of diversity of graphs generated.

**Questions:**

(1) The discrete diffusion loss could contain more variational lower bound terms than cross-entropy. Why do the authors not include the additional terms in the loss calculation?

(2) Why do the authors not report the conditional generation evaluation results of PARD and SwinGNN on QM9 and Zinc molecule dataset?

(3) In the ablation study, where the authors present the results without semi-autoregressive generation, what is the noise scheduling method and how to convert the semi-autoregressive into generate the whole DAG at once?

(4) For molecule generation, if the authors have already converted the molecular structures into DAGs using the junction tree representation, do this conversion losses information of molecules and lead to potentially downgraded performances?

---

### Note · Authors · 2024-11-21

I have read and agree with the venue's withdrawal policy on behalf of myself and my co-authors.